# Structural analysis of the SARS-CoV-2 methyltransferase complex involved in RNA cap creation bound to sinefungin

Petra Krafcikova[1], Jan Silhan[1], Radim Nencka[1✉] & Evzen Boura [1✉]

Severe acute respiratory syndrome coronavirus 2 (SARS-CoV-2) is the cause of the COVID-19 pandemic. 2′-O-RNA methyltransferase (MTase) is one of the enzymes of this virus that is a potential target for antiviral therapy as it is crucial for RNA cap formation; an essential process for viral RNA stability. This MTase function is associated with the nsp16 protein, which requires a cofactor, nsp10, for its proper activity. Here we show the crystal structure of the nsp10-nsp16 complex bound to the pan-MTase inhibitor sinefungin in the active site. Our structural comparisons reveal low conservation of the MTase catalytic site between Zika and SARS-CoV-2 viruses, but high conservation of the MTase active site between SARS-CoV-2 and SARS-CoV viruses; these data suggest that the preparation of MTase inhibitors targeting several coronaviruses - but not flaviviruses - should be feasible. Together, our data add to important information for structure-based drug discovery.

[1] Institute of Organic Chemistry and Biochemistry AS CR, v.v.i., Flemingovo nam. 2., 166 10 Prague 6, Czech Republic. ✉email: nencka@uochb.cas.cz; boura@uochb.cas.cz

Severe acute respiratory syndrome coronavirus 2 (SARS-CoV-2) has caused the coronavirus disease (COVID-19) global pandemic[1], which has currently led to more than 10 million confirmed cases and more than 500 thousands deaths in over 200 countries according to the World Health Organization (www.who.int). Coronaviruses have long been a threat, but recent developments show that they should be classified as extremely dangerous pathogens and that we must develop effective means to suppress and treat the diseases caused by these viruses[2]. Currently, the arsenal of approved treatments for diseases caused by coronaviruses is rather limited and therefore there is a pressing need for the discovery and development of therapeutic agents for treatment of COVID-19 and other coronavirus infections[3]. Directly acting antiviral agents provide a backbone for treatment of numerous viral disease such as hepatitis B and C and AIDS[4,5] and such a compound, remdesivir, was also very recently FDA approved for emergency treatment of COVID-19 patients. These therapeutics directly aim at a certain viral protein and, therefore, a deeper understanding of the function of individual viral proteins is needed to derive future therapies of COVID-19 and other possible coronavirus infections.

Coronaviruses have the largest genomes of all RNA viruses. In particular, the genome of SARS-CoV-2 has ~29 800 bases, which encodes 4 structural and 16 nonstructural proteins (nsp1–nsp16) that are essential for the lifecycle of this virus[6,7]. SARS-CoV-2 exploits the cell environment to its full advantage for its use and replication[8]. Importantly, viral RNA must be protected from the cellular innate immunity. One of the most important elements that ensures the integrity of viral RNA is the cap, a specific arrangement at the 5 'end of the RNA molecule that consists of a *N*-methylated guanosine triphosphate and *C2′*-O-methyl-ribosyl-ladenine (type 1 cap, Fig. 1). This arrangement resembles the native mRNA of the host cells, stabilizes the RNA, and ensures effective process of its translation[9–11]. In human cells, however, the cap is installed on newly transcribed mRNA already in the nucleus, to which coronaviruses do not have access. Instead, they possess their own cap-synthesizing enzymes. Clearly, this process is essential for the survival and further replication of viral RNA in cells. In principle, four different processes are necessary for installation of a type 1 cap on RNA (either human mRNA or coronavirus RNA). First, the γ-phosphate from a 5′-triphosphate end of the nascent RNA is removed by 5′-RNA triphosphatase. Second, a guanosine monophosphate (GMP) is attached to the formed 5′-diphosphate end of RNA by a guanylyltransferase using GTP as the source of GMP. Finally, the methylation steps take place. In this case, two separate enzymatic steps are required: one for *N*-7 methylation of the GTP nucleobase (*N*-7

methyltransferase) and the other for *C2′*-O methylation of the following nucleotide.

Coronaviruses use sequence installation of the cap that is performed by several nonstructural proteins (nsp) encoded by their genome. For coronaviruses, nsp10, 13, 14, and 16 appear to be involved in this process[12]. The primary function of nsp13 is the unwinding of the viral RNA during replication. Therefore, it is considered to be essentially the helicase. However, it is also a protein with 5′-RNA triphosphatase activity responsible for cleaving monophosphate at the 5′-end of the nascent RNA to provide a diphosphate[13]. There is still no clear evidence that any of the coronavirus proteins possess the guanylyltransferase functionality associated with the cap creation[12]. Nsp14 and nsp16 are responsible for the methylation of the cap on the guanine of the GTP and the *C2′* hydroxyl group of the following nucleotide, respectively. Both nsp14 and nsp16 are S-adenosylmethionine (SAM)-dependent methyltransferases (MTases) and seem to be essential for the viral lifecycle[7]. In particular, nsp16 appears to be a very promising molecular target from the perspectives of medicinal chemistry and drug design. It has been shown that this 2′-O methyltransferase (MTase) is indispensable for replication of coronaviruses in cell cultures[14,15]. Enzymatic activities of both these MTases (nsp14 and nsp16) are significantly enhanced by nsp10, which is a necessary cofactor for their proper function[15–19].

Here we report on the crystal structure of SARS-CoV-2 nsp10-nsp16 in complex with sinefungin, a pan- MTase inhibitor originally isolated from *Streptomyces griseoleus*[20]. The structure reveals an overall fold similar to SARS-CoV nsp10-nsp16, and, importantly, reveals atomic details in how sinefungin inhibits the nsp16 MTase. This provides the starting point for specific inhibitor design.

## Results

**Overall structure of the nsp10-nsp16 protein complex.** To obtain the nsp10-nsp16 protein complex we co-expressed the nsp10 and nsp16 encoding genes together in *E. coli*. The complex was stable during protein purification suggesting suitability for structural analysis (Supplementary Fig. 1). The nsp10-nsp16 complex was supplemented with the pan-MTase inhibitor sinefungin and subjected to crystallization trials. Eventually we obtained crystals that diffracted to 2.4 Å resolution. The structure was solved by molecular replacement and revealed a mixed alpha-beta fold with sinefungin bound in a central canyon (Fig. 2). A central feature of the nps16 MTase is a strip of parallel and anti-parallel β-sheets (as they appear in the structure from the nsp10 interface: β4, β3, β2, β6, β7, β9, β8, β1) in the shape of the letter J which is stabilized from the inside by surrounding helices α3 and α4 and from the outside by helices α5-α9.

Nsp10 could be divided into two subdomains: a helical α-subdomain composed of helices α1-α4 and α6 and a β-subdomain composed of two anti-parallel β-sheets (nsp10 β1 and β2), a short helix α5 and several coiled-coil regions. A key feature of the nsp10 fold are two zinc binding sites. One is formed by three cysteine residues (Cys74, Cys77, Cys90) and a histidine residue (His83) and is located between the helices α2 and α3 and appears to stabilize them in the observed conformation. The other zinc binding site is formed by four cysteine residues (Cys117, Cys120, Cys128, Cys130) and stabilizes the very C-terminus of the nsp10 protein.

The nsp10-nsp16 dimer interface is 1983 Å$^2$ large and it is formed by the nsp10 helices α2, α3, α4 and a coiled-coil region connecting helix α1 and the sheet β1 (residues Asn40 to Thr49) and the inner side of the nsp16 J-motif including sheets β4 and helices α3, α4, and α10 (Fig. 3). Nsp10 and nsp16 interact

**Fig. 1 RNA cap.** Methylation performed by nsp14 is highlighted in blue, methylation performed by nsp16 in red. B = base.

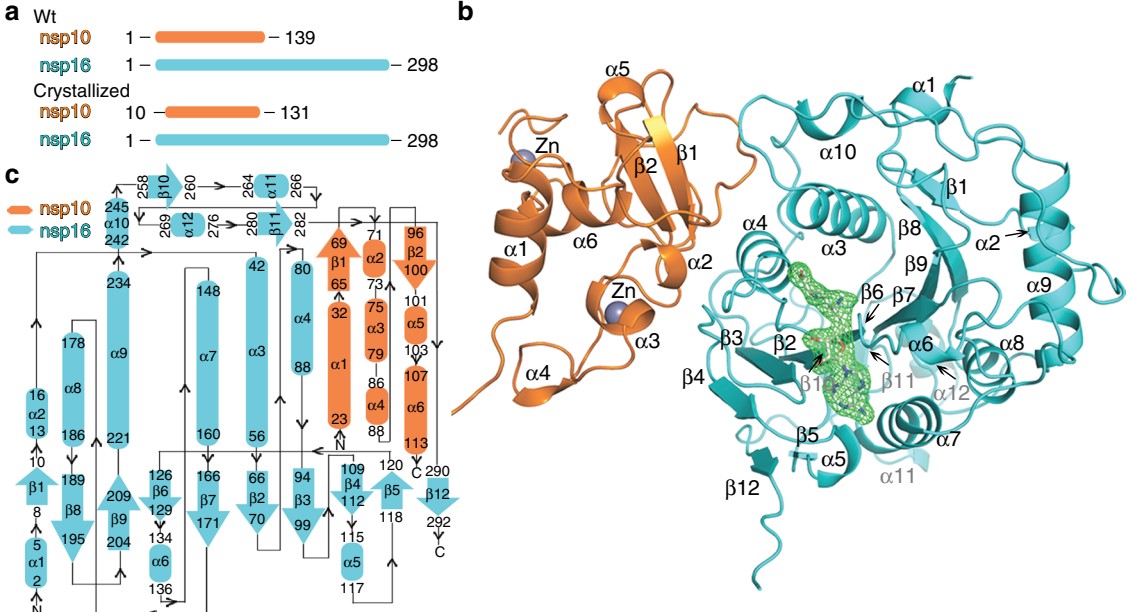

**Fig. 2 Crystal structure of the SARS-CoV-2 nsp10-nsp16 protein complex. a** Schematic comparison of the crystallized and wild-type (wt) nsp10 and nsp16 proteins **b** Overall fold of the nsp10-nsp16 complex with sinefungin bound. The protein backbones are shown in the ribbon presentation. Nsp10 is shown in orange and nsp16 in cyan. Sinefungin is depicted in an unbiased Fo-Fc map contoured at 3.5 sigma. **c** Topology plot of nsp10 and nsp16 proteins.

through a large network of hydrogen bonds often mediated by water molecules (Supplementary Fig. 2) or through hydrophobic interactions. Two residues Val42 and Leu45 of nsp10 are immersed into hydrophobic pockets formed by helices α3, α4, and α10 of nsp16 (Fig. 3e). Nsp10 Val42 is anchored in an nsp16 hydrophobic pocket formed by residues Met41, Val44, Ala73, Val78, and Pro80. Similarly, Leu45 is anchored in a deep hydrophobic pocket formed by nsp16 residues Pro37, Ile40, Val44, Thr48, Leu244, and Met247. Further on, the main chain of the aforementioned nsp10 Leu45 also participates in two hydrogen bonds. The carbonyl group of Leu45 hydrogen bonds with Glu87 of nsp16 directly, while the amine group of the mainchain of Leu45 is connected with residue Thr48 of the nsp16 via water bridge mediated by hydrogen bonds (Supplementary Fig. 2). Among other interactions, a positively charged Lys93 of nsp10 forms three hydrogen bonds with two water molecules (waters #54 and #170). They bridge nsp16 with further hydrogen bonds with the side chain of nsp10 Thr106 and the main chain carbonyl group of nsp10 Ala107 and amine group of Ser105 (Supplementary Fig. 2). This large and complex interface explains the observed stability of the nsp10-nsp16 complex.

**Crystallographic analysis of nsp16-ligand interaction.** Electron density for sinefungin was clearly visible upon molecular replacement. Sinefungin is bound in the SAM binding pocket that is localized in a canyon within nsp16 (Fig. 4). The nsp10 adjacent side is formed by ends of the parallel sheets β2, β3, and β4 and helices α5 and α4. The opposite site of the ligand binding canyon is formed by the sheet β6 and helices α3 and α6. The ligand binding site could also be divided into nucleoside and amino acid (methionine) binding pockets. The nucleoside forms hydrogen bonds with several residues including Asp99, Asn101, and Asp114 while the amino acid part is recognized by Asn43, Asp130, and Lys170 (Fig. 4b). These six residues are absolutely conserved among SARS-CoV-2, SARS and MERS (Supplementary Fig. 3) highlighting the importance of RNA methylation for coronaviruses.

**Analysis of the RNA binding site.** We were also interested in the possible binding mode of the RNA substrate. We analyzed the electrostatic surface potential to reveal a putative RNA binding site. A positively charged canyon proximal to the SAM binding pocket could be easily spotted (Fig. 5a). To figure out the orientation of RNA in the binding pocket we took advantage of the existing structure of human mRNA (nucleoside-2′-O-)-methyltransferase and the Dengue NS5 MTase that were crystallized with a short piece of RNA (PDB codes: 4N48, 5DTO)[21,22]. This structure can be superposed with our structure (Supplementary Fig. 4) to elucidate the RNA binding mode. According to the model, the first nucleoside, 7N-methylated guanosine is bound in the upper part of the RNA binding canyon while the second nucleoside is positioned in the central part of the canyon in a way that its ribose ring gets in close proximity to the amino group of sinefungin, which in this case represents the methyl group to be transferred structurally explaining how nsp16 performs 2′-O methylation (Fig. 5). We approximate – a real crystal structure is necessitated to obtain atomic details of the methylation reaction.

**Discussion**
The size of the RNA genome of coronaviruses is limited by many factors, e.g., the (in)stability of the RNA, fidelity of the RdRps, and its ability to correct excessive mutations and by the limited space for a nucleic acid within the capsid. Therefore, every viral enzyme is a potential drug target because RNA viruses do not have the luxury to encode nonessential accessory proteins. In this study, we have structurally analyzed SARS-CoV-2 2′-O-ribose methyltransferase, an essential enzyme involved in RNA cap formation which ensures stability of the viral RNA because non-methylated RNA located in cytoplasm is prone to degradation and cannot be efficiently translated. Our analysis revealed overall fold similarity between SARS and SARS-CoV-2 nsp10-nsp16 complexes (RMSD = 0.747) and also a high conservation of the SAM binding site among coronaviruses, in fact, all the residues that are involved in ligand-hydrogen bonding are absolutely conserved between SARS, SARS-CoV-2, and MERS. The cocrystal

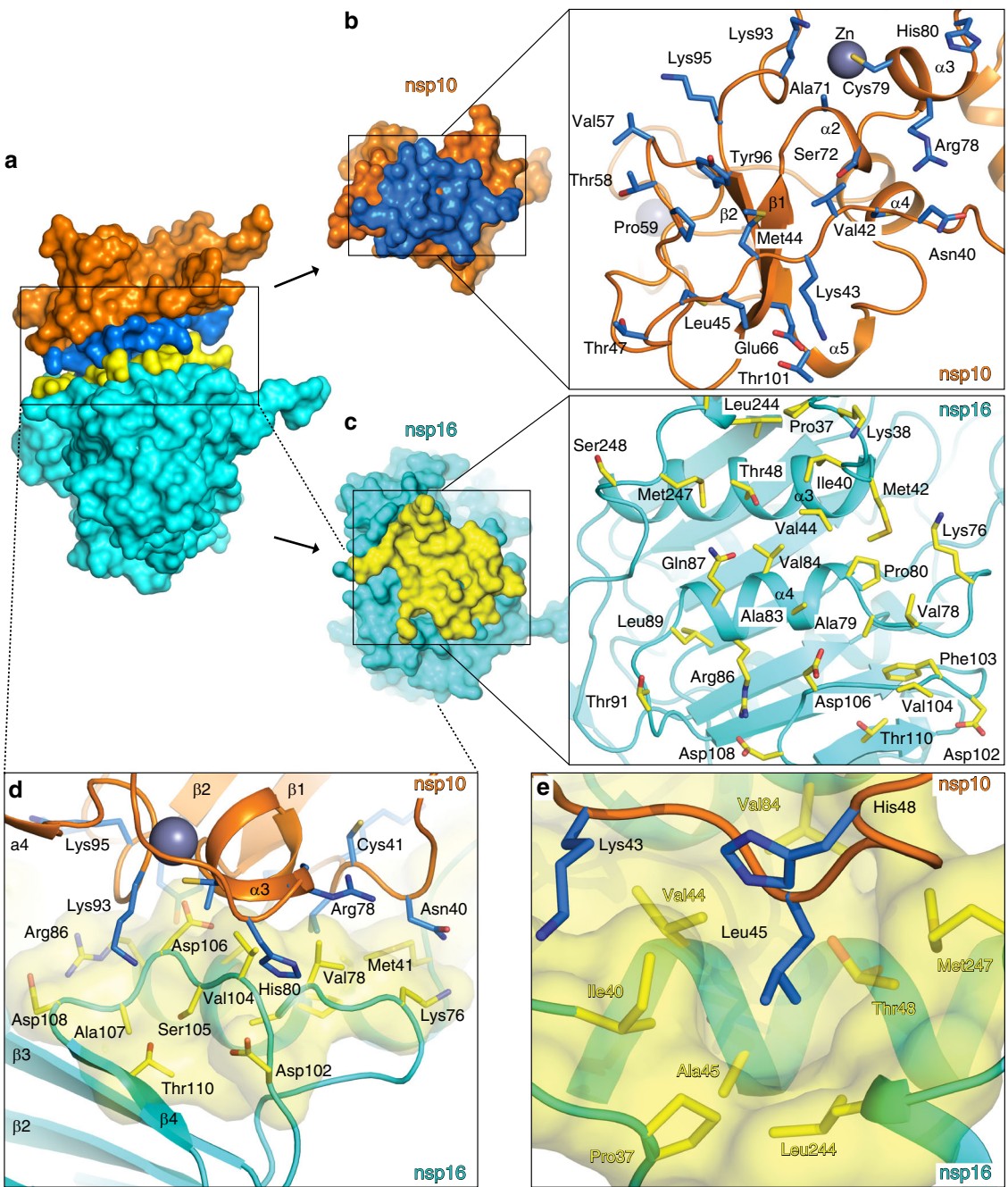

**Fig. 3 Interface of the nsp10-nsp16 protein complex. a** Surface representation of the nsp10-nsp16 protein complex where the interface is labeled in blue (nsp10) and yellow (nsp16). **b** Bottom view of the nsp10 interface; box provides better detail. The interface residues of nsp10 are depicted in blue stick representation. **c** Top view of the nsp16 interface in yellow, where all interface residues are depicted in yellow and shown in greater detail in the box. **d** Side view of the nsp10-nsp16 interface, further details are shown in Supplementary Fig. 2. **e** Residue Leu45 immersed into the hydrophobic pocket defined by interface residues Pro37, Ile40, Val44, Ala45, Thr48,Val84, Leu244, and Met247 of nsp16 (illustrated in partially transparent interface representation in yellow). Involvement of solvent molecules is shown in Supplementary Fig. 2.

structure we obtained is, however, not with the natural methyl donor, SAM, but with the pan-MTase inhibitor sinefungin.

Interestingly, the nps16 MTase is not active without the accessory protein nsp10. The mechanism of nsp16 activation is elusive because we do not have a structure of any unliganded coronaviral nsp16 protein. Structural analysis of the SARS-CoV nsp10 protein in an unliganded form and in complex with nsp16 reveals no significant conformational change of the nsp10 upon nsp16 binding[23]. Therefore, it is expected that nsp10 induces a conformational change in the nsp16 MTase that switches nsp16

in a productive enzyme, although the direct evidence is missing so far. In principle, the nsp10-nsp16 interface is a drug target. However, given the large area of this interface and complex network of hydrogen bonds and hydrophobic interactions it is unlikely that it could be targeted by a small drug-like molecule.

The structure of the nsp10-nsp16 complex reveals several important factors that can be exploited to target the installation of a viral cap on a nascent viral RNA molecule in therapeutic design. The nsp10-nsp16 complex must be able to bind the previously introduced methylated GTP, and recognize at least the first

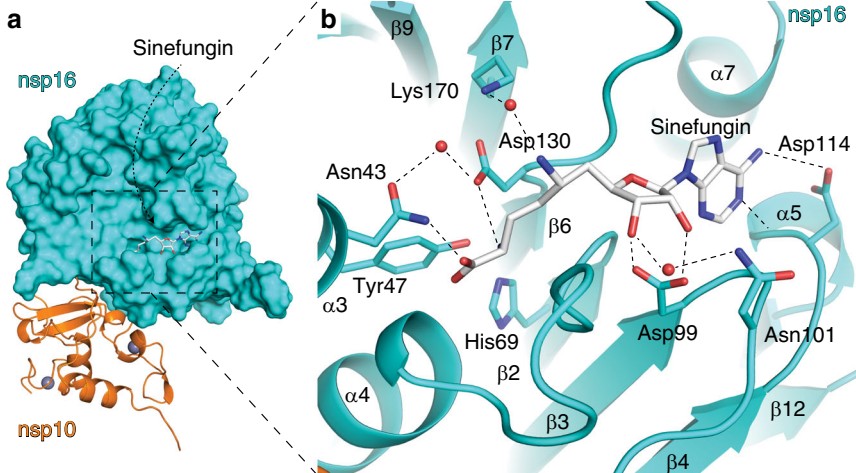

**Fig. 4 Sinefungin recognition by the nsp16 MTase. a** SARS-CoV-2 nsp10-nsp16 protein complex bound to sinefungin (white sticks), nsp16 in surface representation (cyan), nsp10 in cartoon representation (orange), and zinc ions as gray spheres. **b** Detailed view of sinefungin recognition, important amino acid residues are shown in stick representation, waters as red spheres, and hydrogen bonds are shown as dashed lines.

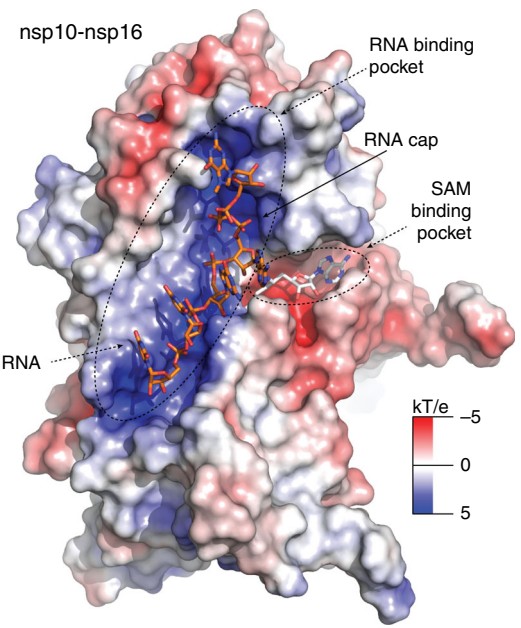

**Fig. 5 Model of RNA recognition by the nsp10-nsp16 complex.** The surface of the nsp10-nsp16 was colored according to the electrostatic surface potential. Sinefungin is localized in the SAM binding pocket. The RNA binding pocket is characterized by a positively charged surface that interacts with the RNA phosphate backbone. The RNA cap is located at the top of the RNA binding pocket while the active site is located at the interface of the RNA and SAM binding pockets. The RNA was modeled in the RNA binding pocket based on a structural alignment of SARS-CoV-2 MTase with Dengue MTase (PDB code 5dto) and manually adjusted in Coot.

nucleotide of the RNA strand, as well as the substrate for the methylation reaction, SAM. Based on the position of sinefungin in our structure, it is apparent where both of these two reaction partners bind to nsp16. These two binding sites form well-defined canyons in the structure of nsp16 as seen in Fig. 5. The binding site for sinefungin must be in direct contact with the 2′-hydroxyl group on the first nucleotide following the introductory methylated GTP moiety. In particular, the chiral amino functionality at C6′ of sinefungin has to be directed toward the RNA cap binding

site. Therefore, modifications on this part of the molecule, based on a rational structure design, has led to the preparation of highly active inhibitors of various MTases as shown previously on various sinefungin-related derivatives[24].

Also detailed knowledge of the amino acid moiety on the sinefungin binding site is extremely important for design of potential SARS-CoV-2 therapeutics, since the possible design of specific bioisosteres of the amino acid scaffold may play a vital role in the development of cell permeable compounds as potential inhibitors of this essential MTase.

We next sought to determine whether an inhibitor of a SERS-CoV-2 MTase could be potentially active against other viruses. We performed a structural alignment of the SARS-CoV-2 MTase with SARS-CoV MTase (Fig. 6a)[25] and with the Zika virus (ZIKV) MTase (Fig. 6b)[26]. The structural comparison reveals high conservation of the MTase active site between the SARS-CoV and the SARS-CoV-2 but rather low conservation of this site between the ZIKV and SARS-CoV-2 MTases. These findings illustrate that development of an MTase inhibitor active against multiple coronavirus species should be feasible, however, development of an MTase inhibitor against both corona- and flaviviruses is rather unlikely unless it would be a promiscuous inhibitor closely resembling the substrate such as sinefungin. However, two residue pairs are conserved among corona- and flaviviruses: (i) the Zika Asp114 and CoV-2 Asp131 that make a hydrogen bond with the adenine base which is essential for its recognition and (ii) Zika Asp130 and CoV-2 Asp146 that are close to the methylation reaction center and important for catalysis. Also, this shows that there is a significant lipophilic cavity in close proximity to the adenine nucleobase of sinefungin in both SARS-CoV-2 and ZIKV MTase. In the case of flavivirus MTases, it has been shown, that this part of the enzyme can be effectively targeted by MTase inhibitors without affecting human proteins. Therefore, we believe that this part of the nsp16 protein may play a very important role in future design of COVID-19 therapeutics. Although there is still a long way to go, through preclinical and clinical testing of inhibitors before they can be introduced into clinical practice, we believe this research provides a solid foundation.

In conclusion, we have acquired a crystal structure of SARS-CoV-2 nsp10-nsp16 complex, the activated 2′-O-methyl-transferse, which is essential for RNA capping during the viral cycle. Since this process is essential for viral survival and

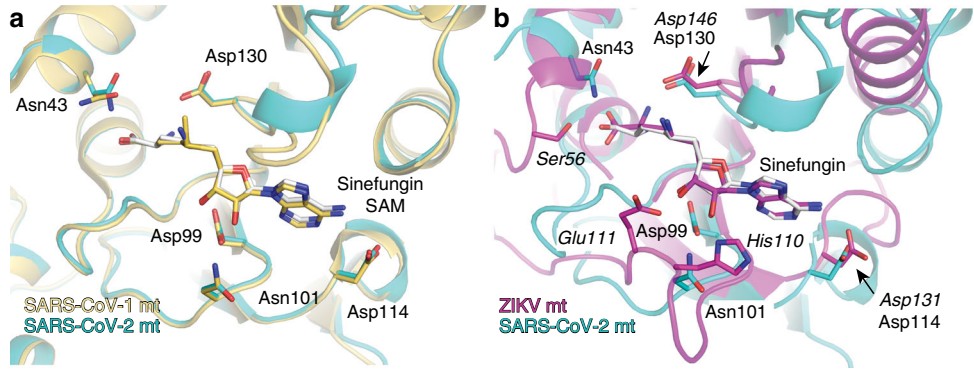

**Fig. 6 Structural alignment of MTase active sites from SARS-CoV-2 with SARS-CoV and ZIKV. a** SARS-CoV-2 and SARS-CoV MTase. **b** SARS-CoV-2 and ZIKV MTase. SARS-CoV-2 nsp16 is shown in cyan, SARS-CoV nsp16 in beige, and ZIKV MTase in magenta. Conserved pairs of residues SARS-CoV-2 Asp114 & ZIKV Asp131 and SARS-CoV-2 Asp130 & ZIKV Asp146 are highlighted by arrows.

### Table 1 Statistics of crystallographic data collection and refinement.

| Crystal | SARS-CoV-2 nsp10-nsp16 |
| --- | --- |
| PDB accession code | 6YZ1 |
| Space group | P3$_1$21 |
| Cell dimensions – a, b, c (Å) | 168.5, 168.5, 52.1 |
| Resolution range (Å) | 48.64 – 2.4 (2.486 - 2.4) |
| No. of total reflections | 881,775 (69,855) |
| No. of unique reflections | 33,374 (3318) |
| Completeness (%) | 99.93 (100.00) |
| Multiplicity | 26.4 (21.1) |
| Mean I/σ(I) | 13.05 (1.63) |
| CC$_{1/2}$ | 0.995 (0.597) |
| CC* | 0.999 (0.865) |
| R$_{merge}$ (%) | 42.0 (246.9) |
| R-work (%) | 18.05 (25.61) |
| R-free (%) | 20.86 (26.82) |
| R.m.s.d. - bonds (Å)/angles (°) | 0.003/0.63 |
| Average B factors (Å$^2$) | 39.26 |
| B factors (Å$^2$) – protein | 37.13 |
| B factors (Å$^2$) – water | 41.79 |
| B factors (Å$^2$) – sinefungin | 30.88 |
| Clashscore | 7.16 |
| Ramachandran favored/outliers (%) | 96.8/0 |

Statistics of crystallographic data collection and refinement. Numbers in parentheses refer to the highest resolution shell.

replication in cells, it can be targeted by chemical compounds based on this structural information.

## Methods

**Cloning, protein expression, and purification.** Artificial codon optimized genes encoding SARS-CoV-2 nsp10 and nsp16 proteins were commercially synthesized (Invitrogen) and cloned in a pSUMO vector that encodes an N-terminal His$_{8×}$-SUMO tag. The proteins were expressed and purified using our standard protocols[27,28]. Briefly, E. coli Bl21 (DE3) cells were transformed with the expression vector and grown at 37 °C in LB medium supplemented with 25 μM ZnSO$_4$. After OD$_{600 nm}$ reached 0.5, the protein expression was induced by addition of IPTG to final concentration 300 μM and the protein was expressed overnight at 18 °C. Bacterial cells were harvested by centrifugation, resuspended in lysis buffer (50 mM Tris, pH 8, 300 mM NaCl, 5 mM MgSO$_4$, 20 mM imidazole, 10% glycerol, 3 mM β-mercaptoethanol) and lysed by sonication. The lysate was cleared by centrifugation. Next, the supernatant was incubated with NiNTA agarose (Machery-Nagel), extensively washed with the lysis buffer and the protein was eluted with lysis buffer supplemented with 300 mM imidazole. The proteins were dialyzed against lysis buffer and digested with Ulp1 protease at 4 °C overnight. The SUMO tag was removed by a second incubation with the NiNTA agarose. Finally, the proteins were loaded on HiLoad 16/600 Superdex 200 gel filtration column (GE Healthcare)

in SEC buffer (10 mM Tris pH 7.4, 150 mM NaCl, 5% glycerol, 1 mM TCEP). Purified proteins were concentrated to 7 mg/ml and stored at −80 °C until needed.

**Crystallization and structure refinement.** Crystals grew in sitting drops consisting of 300 nl of the protein and 150 nl of the well solution (200 mM NaCl, 100 mM Mes, pH 6.5, 10% w/v PEG 4000) in five days. Upon harvest the crystals were cryo-protected in well solution supplemented with 20% glycerol and frozen in liquid nitrogen. Diffraction data were collected at the home source. The crystals diffracted to 2.4 Å and belonged to the trigonal P3$_1$21 spacegroup. Data were integrated and scaled using XDS[29]. The structure was solved by molecular replacement (nps10-nsp16 complex PDB ID 6W4H as the search model) and further refined in Phenix[30] and Coot[31] to good R-factors (R$_{work}$ = 18.05% and R$_{free}$ = 20.86%) and good geometry as summarized in Table 1.

**Reporting summary.** Further information on research design is available in the Nature Research Reporting Summary linked to this article.

## Data availability

The crystal structure was deposited in the RCSB Protein Data Bank, www.pdb.org under an accession code 6YZ1.

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

## Acknowledgements
The work was supported from European Regional Development Fund; OP RDE; Project: "Chemical biology for drugging undruggable targets (ChemBioDrug)" (No. CZ.02.1.01/0.0/0.0/16_019/0000729). We acknowledge CMS-Biocev ("Diffraction") supported by MEYS CR (LM2018127) and the Academy of Sciences of the Czech Republic (RVO: 61388963). We are grateful to Dr. A. Michael Downey (Max Planck Institute of Colloids and Interfaces) for critical reading of the manuscript.

## Author contributions
P.K. performed all experiments, J.S. interpreted the diffraction data, R.N. and E.B. designed and supervised the project. All authors participated in preparation of the manuscript.

## Competing interests
The authors declare no competing interests.
