## [Peer Review File · Nature Communications]

Reviewers' Comments:

Reviewer #1:

Remarks to the Author:

SARS-COV-2 is a positive-sense RNA virus. The genome has the same structure as host mRNA, with 5'cap and 3'polyA tail. The viral protein nsp16 is an RNA 2'O methyltransferase, responsible for the methylation of the 2'O of the first nucleotide. This function of Nsp16 requires a cofactor nsp10 and is essential for the virus. The authors report the crystal structure of the SARS-COV-2 nsp10-nsp16 complex at 2.4 Å. The complex bound to the pan-MTase inhibitor sinefungin. The quality of the structure seems to be good. In principle, this will be a useful starting point in structural based drug design to combat the COVID-19.

Major issue:

2 papers on the crystal structures of Nsp10-Nsp16 have been published, earlier than the current manuscript. The findings are largely overlapping. Further to that, the highly homologous nsp10-16 from SARS-COV has also been extensively studied structurally and enzymatically. Therefore the novelty of the current study is limited.

The crystal structure of nsp10-nsp16 heterodimer from SARS-CoV-2 in complex with S-adenosylmethionine

Monica Rosas-Lemus, George Minasov, Ludmilla Shuvalova, Nicole L. Inniss, Olga Kiryukhina, Grant Wiersum, Youngchang Kim, Robert Jedrzejczak, Natalia I. Maltseva, Michael Endres, Lukasz Jaroszewski, Adam Godzik, Andrzej Joachimiak, Karla J. F. Satchell

bioRxiv 2020.04.17.047498; doi: <https://doi.org/10.1101/2020.04.17.047498>

Structural Basis of RNA Cap Modification by SARS-CoV-2 Coronavirus

Thiruselvam Viswanathan, Shailee Arya, Siu-Hong Chan, Shan Qi, Nan Dai, Robert A. Hromas, Jun-Gyu Park, Fatai Oladunni, Luis Martinez-Sobrido, Yogesh K. Gupta

bioRxiv 2020.04.26.061705; doi: <https://doi.org/10.1101/2020.04.26.061705>

Furthermore, the discussion section, structural analysis also has room to improve. There are more relevant structures of 2'O-MTase: SAM/SAH: Cap0RNA available at PDB database. SI Figure 2 seems to be more informative than the one on ZIKV MTase. There is also another more relevant structure of Dengue virus NS5-MTase in complex with a cap0 RNA (PDB code: 5DTO).

Minor issues:

1. The manuscript title is not aligned with the results. No result of coronaviral RNA cap creation is shown in the manuscript.
2. Site-directed mutagenesis and MTase assay should be carried out to further characterize the SAM binding site.
3. The authors may further characterize the interface and stability of the nsp10-nsp16 complex.
4. In SI Fig 2 sequence alignment, 4 grey stars could be seen but they are unannotated. Are these sites meant to refer to the absolute conservation sites (D99, N101, D114, N43, D130, and K170) mentioned in the main text (Fig 4, page 5)? Reference for the sequence alignment tool used should be included as well.
5. Typo: Two SI Figure 2 in the Supplementary Information

Reviewer #2:

Remarks to the Author:

The authors report the structure of SARS-CoV-2 nsp10-nsp16 in this work. According to the severe situation of COVID-19 pandemic, the insight into virus lifecycle will provide helpful information to understand the virus and lead further antiviral innovation. Therefore, this work is worthy for publication in Nature Communications. However, this manuscript is not well written. Many statements are not correct or overstated (see my detailed comments). The structure comparison with SARS-CoV nsp10-nsp16 should be provided in Results section with additional figures. And several technical issues should be addressed.

Abstract

1. The first sentence should be thoroughly rephrased.
2. "This MTase is composed of two nonstructural proteins, the nsp16 catalytic subunit and the activating nsp10 protein". This statement is not right. MTase activity is in nsp16. Nsp10 is only a co-factor.
3. "Based on the structural data we built a model of the MTase in complex with RNA that illustrates the catalytic reaction." This does not make sense to be mentioned in abstract section.
4. Upon the comparison with Zika MTase, is there any implication from the comparison with other CoV nsp16-nsp10?

Introduction

5. The first sentence should be rephrased. And the current number of infection and death cases should cite the most recent number from the WHO.
6. "Currently, there is no approved" This is not right. There is no approved antiviral therapeutics, but not "no approved treatments of diseases". Actually, there are several immune therapeutics have been used.
7. "Directly acting antiviral agents have revolutionized the treatment..." This sentence should be rephrased.
8. "In particular, the genome of SARS-CoV-2 has ~29 800 bases..." This statement is not right. Structural protein is not directly related to replication. They are essential for virus lifecycle.
9. "As all positive-sense single-stranded" Remove
10. "This arrangement resembles the native mRNA of the host..." this paragraph is lack of appropriate citations.

Results

11. I am wondering whether the lack of subtitle could be accepted according to Nature Communication policy?
12. "we co-expressed the appropriate genes together" what does "appropriate" mean? This sentence should be rephrased.
13. "The complex was stable during protein" a figure for purification should be provided, at least, in supplementary figure.
14. "Nsp10 could be divided in two subdomains a helical" change to "Nsp10 could be divided in two subdomains: a helical"
15. "The nsp10-nsp16 dimer interface is 1983 Å² large" Å²
16. After this paragraph, it should be necessary to give a comparison between SARS-CoV-2 and other CoV nsp10-nsp16.
17. "This structure can be superposed with our structure (SI Figure 3) accurately enough to elucidate the RNA binding mode." "accurately enough" this statement should be removed.
18. "7N-methylated guanosine is bound" This statement is not correct. This is only a proposed model, but not an experimental structure. is potentially located
19. What is the rmsd between sars-cov-2 nsp16 and 4N48?

Discussion

20. "Coronaviruses have the longest genome among RNA viruses." This sentence is repeated with Introduction. Remove
21. "The size of the RNA genome is limited by the (in)stability of the RNA, fidelity of the RdRps, its ability to correct excessive mutations and by the limited space for nucleic acid within the icosahedral capsid." Statements in this sentence have many errors. Proof-reading is also important. And CoV does not have icosahedral capsid.
22. The comparison of sars-cov and sars-cov-2 nsp10-nsp16 should be moved to Results section with appropriate figures.
23. "it is expected that nsp10 induces a conformational change in the nsp16 MTase that" there is no evidence to support this speculation.
24. "corona- and flaviviruses is highly unlikely" in my opinion, the binding sites in CoV nsp16 and flavivirus MTase are generally conserved, but with difference on contacting residues. "Highly unlikely" is not appropriate.
25. "In conclusion, we have..." I agree that the inhibitor of nsp16 might be further developed as therapeutics, but there are still many questions needs to be solved. the toxicity, the PK, and etc. The authors should aware this in Discussion section.

Figure 1

26. Figure 1: is not necessary to be shown in the main text. Could move to supplementary or extended data.

Figure 2

27. A. this panel is confused. Does the author mean there are several residues cannot be traced in the density? Or they crystallized a truncated protein?
28. B. This panel is generally OK, but the labels are not clear represented.
29. C. The labels of the start and end residues for all secondary structure elements make this panel too crowded for clear representation. The number of residues should be introduced in the legends.

Figure 3

30. The panels in figure 3 are generally ok, but all labels are not clear for readers.
31. "Waters are not shown". Any intermolecular interaction mediated by solvent molecules? Why solvent molecules that mediated interactions are not shown?

Figure 4

32. "SARS CoV-2" should be SARS-CoV-2

Table 1

33. No accession pdb code
34. Sg, P3121 should be italic.
35. No. of all reflections should be provided
36. Number of protein atoms, ligands or solvents should be provided. And the B-factor for each group should be individually provided.

SI Figure 1

37. It is clearly that not all residues shown in A and B participate in hydrogen bond formation and not all hydrogen bonds are shown in this figure (actually they are not necessary to be shown). The title "Detailed hydrogen bonding at the interface" is not correct.

Reviewer #1 (Remarks to the Author):

SARS-COV-2 is a positive-sense RNA virus. The genome has the same structure as host mRNA, with 5' cap and 3' polyA tail. The viral protein nsp16 is an RNA 2'O methyltransferase, responsible for the methylation of the 2'O of the first nucleotide. This function of Nsp16 requires a cofactor nsp10 and is essential for the virus. The authors report the crystal structure of the SARS-COV-2 nsp10-nsp16 complex at 2.4 Å. The complex bound to the pan-MTase inhibitor sinefungin. The quality of the structure seems to be good. In principle, this will be a useful starting point in structural based drug design to combat the COVID-19.

We appreciate that the Expert Referee #1 believes that our structure provides a useful starting point in structural based drug design to combat the COVID-19.

Major issue:

2 papers on the crystal structures of Nsp10-Nsp16 have been published, earlier than the current manuscript. The findings are largely overlapping. Further to that, the highly homologous nsp10-16 from SARS-COV has also been extensively studied structurally and enzymatically. Therefore the novelty of the current study is limited. The crystal structure of nsp10-nsp16 heterodimer from SARS-CoV-2 in complex with S-adenosylmethionine (Monica Rosas-Lemus, George Minasov, Ludmilla Shuvalova, Nicole L. Inniss, Olga Kiryukhina, Grant Wiersum, Youngchang Kim, Robert Jedrzejczak, Natalia I. Maltseva, Michael Endres, Lukasz Jaroszewski, Adam Godzik, Andrzej Joachimiak, Karla J. F. Satchell

bioRxiv 2020.04.17.047498; doi: <https://doi.org/10.1101/2020.04.17.047498>, Structural Basis of RNA Cap Modification by SARS-CoV-2 Coronavirus Thiruselvam Viswanathan, Shailee Arya, Siu-Hong Chan, Shan Qi, Nan Dai, Robert A. Hromas, Jun-Gyu Park, Fatai Oladunni, Luis Martinez-Sobrido, Yogesh K. Gupta bioRxiv 2020.04.26.061705; doi: <https://doi.org/10.1101/2020.04.26.061705>).

We are aware that other laboratories were also successful in crystallization of the SARS-CoV-2 nsp10-nsp16 complex. We have also put our manuscript on the bioRxiv preprint server (doi: <https://doi.org/10.1101/2020.05.15.097980>).

Furthermore, the discussion section, structural analysis also has room to improve. There are more relevant structures of 2'O-MTase: SAM/SAH: Cap0 RNA available at PDB database. SI Figure 2 seems to be more informative than the one on ZIKV MTase. There is also another more relevant structure of Dengue virus NS5-MTase in complex with a cap0 RNA (PDB code: 5DTO).

Excellent suggestion. We have modified Figure 5 to show RNA in the RNA binding pocket. The RNA was modeled based on the Dengue virus NS5-MTase in complex with a cap0 RNA (PDB code: 5DTO). We have also added a new SI Figure showing the superposition of Dengue virus NS5-MTase with our structure. The MTase figure was modified to show the same level of details as the SI Figure 2.

Minor issues:

1. The manuscript title is not aligned with the results. No result of coronaviral RNA cap creation is shown in the manuscript.

The manuscript title was chosen to be informative for a general reader who may not be familiar with coronaviral biology and may not know that coronaviruses modify their RNA with a cap. However, we have modified the title to be more accurate. Now the title of our manuscript is:

Structural analysis of the SARS-CoV-2 methyltransferase complex involved in coronaviral RNA cap creation bound to the pan-methyltransferase inhibitor sinefungin

2. Site-directed mutagenesis and MTase assay should be carried out to further characterize the SAM binding site.

Excellent suggestion. However, it would slow down publication of our manuscript by several months. Given the COVID-19 situation we prefer to publish our structural findings as soon as possible.

3. The authors may further characterize the interface and stability of the nsp10-nsp16 complex.

As above.

4. In SI Fig 2 sequence alignment, 4 grey stars could be seen but they are unannotated. Are these sites meant to refer to the absolute conservation sites (D99, N101, D114, N43, D130, and K170) mentioned in the main text (Fig 4, page 5)? Reference for the sequence alignment tool used should be included as well.

We thank Expert Referee #1 for catching this error. Legend of SI Fig. 2 was changed, stars were removed. The reference for sequence alignment tool was added.

5. Typo: Two SI Figure 2 in the Supplementary Information

We thank Expert Referee #1 for catching this typo. It was corrected.

Reviewer #2 (Remarks to the Author):

The authors report the structure of SARS-CoV-2 nsp10-nsp16 in this work. According to the severe situation of COVID-19 pandemic, the insight into virus lifecycle will provide helpful information to understand the virus and lead further antiviral innovation. Therefore, this work is worthy for publication in Nature Communications. However, this manuscript is not well written. Many statements are not correct or overstated (see my detailed comments). The structure comparison with SARS-CoV nsp10-nsp16 should be provided in Results section with additional figures. And several technical issues should be addressed.

We appreciate that the Expert Referee #2 believes that our work is worthy for publication in Nature Communications.

The text and figures were significantly improved according to suggestions of Expert Referee #2.

Abstract

1. The first sentence should be thoroughly rephrased.
2. “This MTase is composed of two nonstructural proteins, the nsp16 catalytic subunit and the activating nsp10 protein”. This statement is not right. MTase activity is in nsp16. Nsp10 is only a co-factor.
3. “Based on the structural data we built a model of the MTase in complex with RNA that illustrates the catalytic reaction.” This does not make sense to be mentioned in abstract section.
4. Upon the comparison with Zika MTase, is there any implication from the comparison with other CoV nsp16-nsp10?

We agree with the Expert Referee #2 on all four points. We have corrected the abstract accordingly. We also included a new Figure 6A that is comparing the active site of SARS-CoV and SARS-CoV-2 MTase.

Introduction

5. The first sentence should be rephrased. And the current number of infection and death cases should cite the most recent number from the WHO.

We have rephrased this sentence and included numbers of confirmed cases and deaths according to WHO website (7.6 million of confirmed cases and more than 400 thousands deaths). Unfortunately, with the dynamics of the COVID-19 pandemic, these numbers might be outdated soon.

6. “Currently, there is no approved“. This is not right.

There is no approved antiviral therapeutics, but not “no approved treatments of diseases”. Actually, there are several immune therapeutics have been used.

We have changed this sentence to be accurate to: “Currently, the arsenal of approved treatments for diseases caused by coronaviruses is rather limited and therefore...”

7. “Directly acting antiviral agents have revolutionized the treatment”; This sentence should be rephrased.

We have rephrased this sentence to: “Directly acting antiviral agents are a backbone of the treatment...”

- 8.”In particular, the genome of SARS-CoV-2 has ~29 800 bases”; This statement is not right. Structural protein is not directly related to replication. They are essential for virus lifecycle.

Indeed, we apologize for the inaccuracy. Now we say: “..which encodes four structural and sixteen nonstructural proteins (nsp1 - nsp16) that are essential for the lifecycle of this virus.”

9. “As all positive-sense single-stranded“; Remove

We have removed this part of the sentence.

10. “This arrangement resembles the native mRNA of the host” this paragraph is lack of appropriate citations.

We have added an appropriate citation for the paragraph.

Results

11. I am wondering whether the lack of subtitle could be accepted according to Nature Communication policy?

Subtitles were added in the results section.

12. “we co-expressed the appropriate genes together” what does ”appropriate” mean? This sentence should be rephrased.

The sentence was rephrased. Now we say:

To obtain the nsp10-nsp16 protein complex we co-expressed the nsp10 and nsp16 encoding genes together in *E. coli*.

13. “The complex was stable during protein” a figure for purification should be provided, at least, in supplementary figure.

A new SI Figure 1 is included. Now we state:

The complex was stable during protein purification suggesting suitability for structural analysis (SI Fig. 1).

14. “Nsp10 could be divided in two subdomains a helical” change to ”Nsp10 could be divided in two subdomains: a helical”

Changed as suggested.

15. The nsp10-nsp16 dimer interface is 1983 Å² large.

Was changed to:

The nsp10-nsp16 dimer interface is 1983 Å² large.

16. After this paragraph, it should be necessary to give a comparison between SARS-CoV-2 and other CoV nsp10-nsp16.

We now present a new Figure 6A illustrating the conservation of the SARS-CoV-2 and SARS-CoV MTase active sites.

17. "This structure can be superposed with our structure (SI Figure 3) accurately enough to elucidate the RNA binding mode." accurately enough - this statement should be removed.

Removed as suggested.

18. "N-methylated guanosine is bound" This statement is not correct. This is only a proposed model, but not an experimental structure. Is potentially located

Indeed, we apologize. Now we state:

According to the model, the first nucleoside, 7N-methylated guanosine is bound in the upper part of the RNA binding

19. What is the rmsd between sars-cov-2 nsp16 and 4N48?

Actually, in this case PyMol fails in structural alignment based on C-alpha atoms. We performed the structural alignment based on the ligands and therefore, in this particular case, reporting the exact number over C-alpha atoms could be misleading.

Discussion

20. "Coronaviruses have the longest genome among RNA viruses." This sentence is repeated with Introduction. Remove

Removed as suggested.

21. "The size of the RNA genome is limited by the (in)stability of the RNA, fidelity of the RdRps, its ability to correct excessive mutations and by the limited space for nucleic acid within the icosahedral capsid." Statements in this sentence have many errors. Proof-reading is also important. And CoV does not have icosahedral capsid.

We apologize for the inaccurate statement. Now we state:

"The size of the RNA genome of coronaviruses is limited by many factors, e.g. the (in)stability of the RNA, fidelity of the RdRps and its ability to correct excessive mutations and by the limited space for nucleic acid within the capsid. Therefore,..."

22. The comparison of sars-cov and sars-cov-2 nsp10-nsp16 should be moved to Results section with appropriate figures.

We now present a new Figure 6A illustrating the conservation of the SARS-CoV-2 and SARS-CoV MTase active sites. We believe that comparison of SARS-CoV and SARS-CoV-2 structures fits the discussion section better because it has important implication for inhibitor design.

23. “it is expected that nsp10 induces a conformational change in the nsp16 MTase that” there is no evidence to support this speculation.

Indeed, we have added a part of sentence saying:

“..., although the direct evidence is missing so far.”

24. “corona- and flaviviruses is highly unlikely” in my opinion, the binding sites in CoV nsp16 and flavivirus MTase are generally conserved, but with difference on contacting residues. “Highly unlikely” is not appropriate.

Indeed, at least two residues are highly conserve between Zika and SARS-CoV-2. We have changed “highly unlikely” to “rather unlikely”.

25. “In conclusion, we have” I agree that the inhibitor of nsp16 might be further developed as therapeutics, but there are still many questions needs to be solved. the toxicity, the PK, and etc. The authors should aware this in Discussion section.

We are aware of how difficult it is to develop an inhibitor into a drug.

We changed the previous sentence to: “Therefore, we believe that this part of the nsp16 protein may play a very important role in future design of novel COVID-19 therapeutics, although it is clear that there is still a long way, through preclinical and clinical testing of inhibitors, before they can be introduced into clinical practice.”

Figure 1

26. Figure 1: is not necessary to be shown in the main text. Could move to supplementary or extended data.

Here we must disagree with the Expert Referee #2. While the Referee is an expert in coronaviral biology and this figure may not provide much for him/her, it is very informative for a general reader.

Figure 2

27. A. this panel is confused. Does the author mean there are several residues cannot be traced in the density? Or they crystallized a truncated protein?

Nsp10 used for crystallization was truncated.

28. B. This panel is generally OK, but the labels are not clear represented.

The labels were moved and the font increased for clarity.

29. C. The labels of the start and end residues for all secondary structure elements make this panel to crowded for clear representation. The number of residues should be introduced in the legends.

The figure was changed. We enlarged the image not to look crowded.

Figure 3

30. The panels in figure 3 are generally ok, but all labels are not clear for readers.

The labels were changed to bold and moved for better visibility.

31. “Waters are not shown” Any intermolecular interaction mediated by solvent molecules? Why solvent molecules that mediated interactions are not shown?

Water mediated interactions are not shown in Figure 3 for clarity (the nsp10-nsp16 interface is large and complex). However, water mediated contacts and hydrogen bonds are shown in SI Figure 2. Water mediated interactions with the ligand in Figure 4.

Figure 4

32. “SARS CoV-2” should be SARS-CoV-2

The typo was corrected.

Table 1

33. No accession pdb code

We appreciate that the Expert Referee #2 noticed this omission. PDB code was added.

34. Sg, P3121 should be italic.

Changed as suggested, the capital letter P is now in italic.

35. No. of all reflections should be provided.

No. of total reflection was added. Actually we believe it is not necessary because we already state the number of unique reflections and redundancy.

36. Number of protein atoms, ligands or solvents should be provided. And the B-factor for each group should be individually provided.

We believe that these details do not provide any significant insight. Moreover, an interested reader can find them in the PDB database.

SI Figure 1

37. It is clearly that not all residues shown in A and B participate in hydrogen bond formation and not all hydrogen bonds are shown in this figure (actually they are not necessary to be shown). The title “Detailed hydrogen bonding at the interface” is not correct.

We thank the Expert Referee 2 for pointing out this inaccuracy. We agree that not all hydrogen bonds are shown and also with the statement that they are not necessary to be shown. We have changed the title of the SI Figure 1 (now SI Figure 2) to be more accurate. Now we state:

Selected interactions at the vast nsp10-nsp16 interface